# From Jack of All Trades to Master of One: Specializing LLM-based Autoraters to a Test Set

**Mara Finkelstein** [1]  **Daniel Deutsch** [1]  **Parker Riley** [1]  **Juraj Juraska** [1]  **Geza Kovacs** [1]  **Markus Freitag** [1]

## Abstract

As LLMs continue to become more powerful and versatile, human evaluation has become intractable at scale and reliance on automatic metrics has become the norm. Recently, it has been shown that LLMs are themselves state-of-the-art evaluators for many tasks. These *Autoraters* are typically designed so that they generalize to new systems *and* test sets. In practice, however, evaluation is performed on a small set of fixed, canonical test sets, which are carefully curated to measure the capabilities of interest and are not changed frequently. In this work, we design a method which specializes a prompted Autorater to a given test set, by leveraging historical ratings on the test set to construct in-context learning (ICL) examples. We evaluate our *Specialist* method on the task of fine-grained machine translation evaluation, and show that it dramatically outperforms the state-of-the-art XCOMET metric by 54% and 119% on the WMT'23 and WMT'24 test sets, respectively. We also evaluate our method on the task of open-ended story generation evaluation, where we show that it outperforms the non-specialized baseline by 31% on the HANNA benchmark. We perform extensive analyses to understand the representations learned by our Specialist metrics, and how variability in rater behavior affects their performance. We also verify the generalizability and robustness of our Specialist method across different numbers of ICL examples, LLM backbones, systems to evaluate, and evaluation tasks.

## 1. Introduction

While evaluation of natural language generation (NLG) systems has been a long-standing challenge, its importance has come to the fore in the era of large language models (LLMs). Moreover, while human evaluation has historically been considered the gold standard for measuring model quality, it has become a key bottleneck during model development. In addition to being costly, slow, and difficult to scale, human evaluation is also limited by subjectivity (Krishna et al., 2023) and high variability in judgments across human raters (Karpinska et al., 2021; Riley et al., 2024; Zhang et al., 2024b). Increasingly, automatic metrics are replacing human evaluation for measuring the quality of generative models, and LLMs themselves have been shown to be state-of-the-art evaluators (also known as *Autoraters*) across a range of capabilities (Kim et al., 2023; 2024; Vu et al., 2024; Li et al., 2023).

The race to build ever-more-performant LLMs has accelerated not only this shift from human to automatic evaluation, but has also brought demand for standard test sets on which LLM quality is measured and compared (Hendrycks et al., 2020; Liang et al., 2022; Zheng et al., 2023a;b). Evaluating new systems on a fixed set of benchmarks, which are carefully curated to measure certain capabilities of interest and are not changed frequently, allows for fair comparison against previous work and is the standard in the literature. Thus, while automatic metrics are typically designed so that they generalize to new systems *and* test sets, in practice, it is very important that the evaluation metric being used work well across systems on the given test set, and less important that the metric generalize to other, unseen and unused test sets. In this work, we propose a simple and highly effective method to build LLM-based Autoraters which are specialized to a given test set, by leveraging historical ratings on the test set to construct ICL examples.

Our contributions can be summarized as follows:

- We propose a novel method for constructing LLM-based Autoraters for NLG evaluation, which are specialized to a given test set (see Figure 1a). This method only requires multi-shot prompting (no finetuning).

- We show that this method can be used to create an automatic metric for fine-grained machine translation (MT) evaluation (called Specialist AutoMQM) which dramatically outperforms the existing state-of-the-art,

[1]Google. Correspondence to: Mara Finkelstein <marafin@google.com>.

*Proceedings of the 42nd International Conference on Machine Learning*, Vancouver, Canada. PMLR 267, 2025. Copyright 2025 by the author(s).

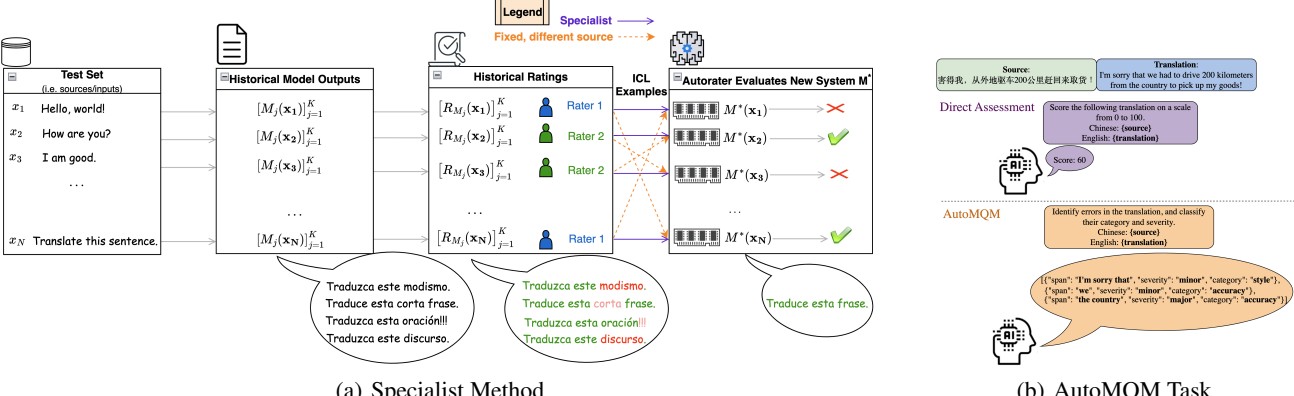

(a) Specialist Method                    (b) AutoMQM Task

*Figure 1.* (a) Illustration of the `Specialist` method, compared against the `Fixed, different source` baseline, for prompting an LLM-based Autorater. Both methods construct a unique set of ICL examples for every test set example, consisting of ratings of historical system outputs for some *fixed source*. The difference between these methods is that the `Specialist` ICL examples consist of ratings of outputs from the *same source* as the test example. (b) Illustration of the AutoMQM task for fine-grained MT evaluation.

achieving character-level F1 improvements of 54% and 119% on the WMT'23 and WMT'24 test sets, respectively, relative to XCOMET (Guerreiro et al., 2023).

- We perform extensive ablations and analyses to verify that the representations that Specialist AutoMQM learns from the ICL examples are non-trivial. We also show that the Specialist method is robust to the choice of LLM and to the systems being evaluated, and that this method generalizes to the different, but related, task of score prediction for machine translation.

- We investigate how variability in judgments across different human raters affects performance of Specialist AutoMQM, and conclude that this metric specializes not only to the test set, but also to the rater.

## 2. Related Work

**LLM-as-a-Judge Autoraters** Recently, it has been shown that, for many NLG tasks, LLMs are themselves state-of-the-art evaluators (Kim et al., 2023; 2024; Vu et al., 2024; Li et al., 2023). Some of these "LLM-as-a-Judge" *Autoraters* are finetuned on human judgements (Kim et al., 2023; 2024; Vu et al., 2024; Li et al., 2023), while others are simply prompted (Kocmi & Federmann, 2023a; Yuan et al., 2023). Prompting LLMs with in-context learning (ICL) examples is a common approach for eliciting their reasoning and instruction-following capabilities (Tanzer et al., 2023; Yan et al., 2023). While traditional automatic metrics predict scalar quality scores, the transition towards generative Autoraters for evaluation opens up the possibility to elicit feedback more flexibly, including fine-grained and interpretable feedback (Fernandes et al., 2023; Kocmi & Federmann, 2023a). However, Kamoi et al. (2024) showed that GPT-4 and Claude-3 have low recall in detecting errors

made by LLMs, and their explanations are unreliable.

**Modeling Rater Behavior** For many (NLG) tasks on which LLMs are evaluated, there is high variability in judgments across human raters (Karpinska et al., 2021; Riley et al., 2024). For some (open-ended text generation) tasks, the evaluation criteria have some degree of subjectivity (Krishna et al., 2023). Especially for expert-level evaluation tasks, differences in rater quality and conscientiousness can manifest as inter-annotator disagreement (Karpinska et al., 2021). Raters can also have different stylistic preferences, and some grade more leniently or harshly than others (Riley et al., 2024). Given these differences in rater behavior, several recent studies have sought to model behavior of multiple raters when designing automatic metrics (Zhang et al., 2024b; Geva et al., 2019; Chen et al., 2024; Golazizian et al., 2024). In this work, we instead investigate whether our proposed method effectively specializes to a single rater. In practice, modeling a single, high-quality rater is often more desirable than modeling multiple, noisy raters.

**Machine Translation Evaluation** In this work, we focus on the task of MT evaluation to investigate the effectiveness of our *Specialist* method. MT is a core NLG task, and automatic MT evaluation is one of the most well-studied evaluation problems in NLP (Callison-Burch et al., 2008; Freitag et al., 2023). In line with broader trends, research in automatic MT evaluation has recently shifted towards the LLM-as-a-Judge paradigm. Kocmi & Federmann (2023b) showed that LLMs prompted to predict scalar scores are state-of-the-art evaluators of MT quality at the system level (though still lag behind finetuned MT evaluation metrics at the segment level). While these score-based metrics have high correlation with human judgments, the scores that they produce are difficult to interpret, and do not provide ac-

tionable insights into the limitations of the model being evaluated or how to improve it (Xu et al., 2024; Zhang et al., 2024a). Recent work in creating interpretable *automatic* MT metrics has built upon an existing, state-of-the-art framework for interpretable *human* evaluation of translation quality: the Multidimensional Quality Metrics (MQM; Lommel et al., 2014; Freitag et al., 2021) framework, in which professional annotators are asked to identify and label individual error spans in MT outputs, along with the corresponding error category (e.g., *fluency*, *accuracy*, etc.) and severity (*minor*, *major*, or *critical*). Fernandes et al. (2023) and Kocmi & Federmann (2023a) showed that LLMs can be few-shot prompted to provide MQM error annotations of MT outputs. However, these prompted Autoraters still underperform (Freitag et al., 2023) XCOMET (Guerreiro et al., 2023), an encoder model finetuned on human-generated MQM data, which predicts both scalar quality scores (with a regression head) and error spans (non-generatively) via token-level tagging.

## 3. *Specialist* Method

In this work, we propose the *Specialist* method for development of a prompted LLM-as-a-Judge metric, which specializes the metric to a given test set based on the ICL examples provided. This method will be phrased in terms of the MT evaluation task, but its formulation generalizes to any NLG evaluation task (i.e., any task which evaluates output from generative models).

**Prerequisites**  First, we establish some basic terminology. The objective is to evaluate the performance of an MT system (i.e., model) $M$ on a fixed *test set*. A test set simply consists of a set of *sources* $X$, which are the inputs to the system(s) to be evaluated. (In this setting, we do *not* require access to gold reference translations of these sources.) The *translations* $Y_M$ are the outputs of $M$ on the test set: that is, $Y_M = \{M(x) : x \in X\}$. Evaluation of system $M$ on the test set (whether by MQM, AutoMQM, score prediction, etc.) produces a set of *ratings* $R_M = \{\text{rating}(y) : y \in Y_M\}$ for the translations $Y_M$.

**Specialist Algorithm**  The Specialist method can be summarized as follows: Given access to a test set $X$ augmented with historical translation quality ratings from multiple systems, and given the predictions of a new translation system $M^*$ which we want to evaluate on this test set, the Specialist metric evaluates the quality of $M^*(x)$ for every $x \in X$ by prompting an LLM with ICL examples constructed from all ratings of historical translations of the same input $x$. See Figure 1a for an illustration of the Specialist method.

More formally, Algorithm 1 outlines the details of this method, which requires access to multiple (historical) sets

of (human-generated) ratings $R_{M_j}$ (from different translation systems $M_j$) on the same test set. The *pseudo-SxS* setting primarily considered in this work has the additional requirement that, for each test set example $x_i$, all ratings $\left\{R_{M_j}[i]\right\}_{j=1}^N$ be performed by a fixed rater. In this work, the different translations for each input example come from different translation systems, but they could in principle also be sampled from a single model (e.g., using a diversity-promoting sampling algorithm). The Specialist method constructs ICL examples to be used for prompting an LLM-as-a-Judge on a per-example basis, so that ICL examples are unique for every example in the test set. In particular, for a given input $x_i$ in the test set, the ICL examples are constructed from all of the (historical) ratings of translations of this same input (line 9 in Algorithm 1). That is, given a new translation system $M^*$ to evaluate on the test set, the ICL examples used to evaluate the translation $Y_{M^*}^i = M^*(x_i)$ are given by $\left\{R_{M_j}[i]\right\}_{j=1}^N$. Once the ICL examples are constructed, the LLM is prompted with these demonstrations, as well as the corresponding source $x_i$ and model translation $Y_{M^*}^i$ to evaluate (line 13 in Algorithm 1).

---

**Algorithm 1** Specialist Method for Automatic Evaluation

**Given:**
1: Test set $X = \{x_i\}_{i=1}^K$
2: Translation system $M^*$ to evaluate

**Require:**
3: Off-the-shelf LLM $E$ to use as the prompted Autorater
4: Set $R$ of ratings on $X$ for $N$ translation systems: $R = \left\{R_{M_j}\right\}_{j=1}^N$, where $M_j \neq M^*$ for all $j \in \{1, \dots, N\}$.
   Pseudo-SxS Constraint: For each $i$, ratings $\left\{R_{M_j}[i]\right\}_{j=1}^N$ were performed by a single rater.

**Ensure:** Ratings $R_{M^*}$ of system $M^*$ on test set $X$
5: $R_{M^*} \leftarrow []$
6: # Iterate over examples in the test set
7: **for** $i \leftarrow 1$ to $K$ **do**
8:    # Construct ICL examples from all historical ratings for the same input $x_i$
9:    $(\texttt{ICL examples})_i = \left\{R_{M_j}[i]\right\}_{j=1}^N$
10:   # Compute output of $M^*$ on this test set example
11:   $Y_{M^*}^i = M^*(x_i)$
12:   # Prompt $E$ to evaluate the translation of the new system $M^*$ of input $x_i$, given the historical ratings
13:   $R_{M^*}^i = E\Big((\texttt{ICL examples})_i, x_i, Y_{M^*}^i\Big)$
14:   Append $R_{M^*}^i$ to $R_{M^*}$
15: **end for**
16: **Return** $R_{M^*}$

---

**Specialist Method in Practice**  The main constraint in development of a Specialist metric is the availability of ratings to use as ICL examples for the given test set. However, note that it is much more efficient to collect a set of ratings from a

few translation systems for a test set as a one-off investment, than to repeatedly depend on human annotators for evaluation of new translation models (e.g., throughout the model development process). Performance of Specialist metrics as a function of the number of ratings will be explored in §5.3, where we show that ratings from only 3 translation systems are sufficient to exceed the state-of-the-art.

# 4. Experimental Setup

## 4.1. Evaluation Task: AutoMQM

In this work, we develop a Specialist metric for MT evaluation based on the Multidimensional Quality Metrics (MQM; Lommel et al., 2014; Freitag et al., 2021) protocol (see Section 2). We refer to any automatic evaluation metric which performs the task of MQM as *AutoMQM*. That is, an AutoMQM metric predicts error spans, and identifies corresponding error categories and severities, according to the MQM framework (Figure 1b). In §5.6, we also evaluate our proposed *Specialist* method on the task of MT evaluation via direct assessment (scalar score prediction). The instructions used to prompt our AutoMQM and direct assessment metrics are shown in Figures 5 and 6 in Appendix A, respectively. The AutoMQM prompt was adapted from the GEMBA instructions (Kocmi & Federmann, 2023a). As indicated in the prompt, the output (and ICL examples) are expected to be provided in JSON format, with each error having `span`, `severity`, and `category` fields. See Table 7 in Appendix A for an example AutoMQM output.

## 4.2. Models

We use the Gemini 1.5 Pro model (Gemini Team, 2024) as the prompted LLM Autorater for all experiments (unless otherwise indicated; see §5.1).

### 4.2.1. BASELINES

We compare our proposed Specialist AutoMQM metric against the following baselines:

**XCOMET** (Guerreiro et al., 2023) State-of-the-art automatic metric for *span-based* MT evaluation.

**GEMBA-MQM** (Kocmi & Federmann, 2023a) Closest precedent to our proposed metric, which also prompts an LLM (GPT-4) for the task of MQM prediction. GEMBA uses a fixed set of 3 (English-German, English-Czech, and Chinese-English) ICL examples.

**MetricX** (Juraska et al., 2024) State-of-the-art automatic metric for MT evaluation and winner of the WMT'24 Metrics Shared Task (Freitag et al., 2024). This model is used as a baseline for the Specialist Scorer experiments in §5.6.

**Shuffled sources** The same global set of ICL examples (per test set) as the Specialist model is used, but these ICL examples are shuffled across test examples.

**Fixed, different source** The same global set of ICL examples (per test set) as the Specialist model is used, but these are permuted so that, for a given test example, its ICL examples come from a fixed source, which is strictly different than that of the test example, but has the same rater. See Figure 1a for an illustration of this setup vs the Specialist.

For the "Shuffled sources" and "Fixed, different source" baselines, the following constraint is enforced: The ICL examples for a given test example cannot include any translations, whether from the same or different source, produced by the same system as that which produced the test translation. Moreover, both of these baselines use the same number of ICL examples per test example as the Specialist model.

## 4.3. Test Sets

The Specialist method (described in §3) depends on having access to a test set augmented with multiple ratings (of different translations) for each input. Such ratings have already been collected as part of the Conference on Machine Translation (WMT) Metrics Shared Tasks in 2023 (Freitag et al., 2023) and 2024 (Freitag et al., 2024). We will refer to these datasets as WMT'23 and WMT'24, respectively. We use the WMT'23 MQM ratings for English-German (en→de) and Chinese-English (zh→en), and the WMT'24 MQM ratings for English-German (en→de), English-Spanish (en→es), and Japanese-Chinese (ja→zh). We exclude the human-generated references, so that our metrics are reference-free (i.e., QE). See Table 8 in Appendix B for the number of translation systems per language pair. Except for the en→es WMT'24 dataset, all ratings were collected in a *pseudo-SxS* fashion (Riley et al., 2024), which means that a fixed rater was assigned to rate all translations of a given input.

### 4.3.1. ADDITIONAL ROUNDS OF RATINGS

In order to better understand how inter-rater variability affects the performance of the Specialist metric, we take advantage of additional rounds of MQM ratings (see §5.5).

**WMT'23 Round2 and Round3** Two additional rounds of WMT'23 MQM ratings, rated by the same set of raters as in the first round, but with individual translations being assigned to strictly different raters in each round. As with the first round, the second two rounds of ratings were also collected in a pseudo-SxS fashion.

**WMT'23 Multi-Rater Subset** An extension to the (Round1) WMT'23 zh→en MQM ratings, whereby 10% of the test set (18 source segments × 15 systems = 270 examples) was rated by all 8 raters.

| Character-level F1 | Same source as test | Fixed source | Same rater | WMT'23 | WMT'24 |
|---|---|---|---|---|---|
| **Baselines** | | | | | |
| 1a) XCOMET-XXL-QE | — | — | — | 33.50 | 16.23 |
| 1b) GEMBA-MQM-QE | ✗ | ✗ | ✗ | 31.99 | — |
| 1c) Shuffled sources | ✗ | ✗ | ✗ | 34.65 | 24.46 |
| 1d) Fixed, different source | ✗ | ✓ | ✓ | 27.06 | 19.96 |
| 2a) Specialist | ✓ | ✓ | ✓ | **51.59**$^*$ | **35.59**$^*$ |

*Table 1.* Character-level F1 on the WMT'23 and WMT'24 test sets, for Baseline and Specialist metrics (see §4.2 for metric descriptions), averaged over all language pairs per test set. See Table 11 in Appendix B for F1, precision, and recall, broken out by language pair. For WMT'23, the "Shuffled sources" and "Fixed, different source" results are computed as the average over 10 runs with different random seeds. See Table 12 in Appendix B for the variance across runs. Asterisks ($*$) indicate scores which are statistically significantly better than XCOMET (row 1a), according to a paired permutation test.

## 4.4. Creation of Specialist ICL examples

The Specialist metric (as described in §3) evaluates the quality of a single translation system $M^*$ given known ratings for a set of $N$ other translation systems $\{M_j\}_{j=1}^N$. Here, for each test set (WMT'23 and WMT'24), we have access to ground-truth ratings for all system outputs. In order to meta-evaluate the Specialist AutoMQM metric, we first collect predictions from this metric for each system, via hold-one-out prompting; that is, for whichever system we are evaluating, we exclude that system's ratings from the ICL examples and prompt with the ratings from the remaining systems. Then, we gather the Specialist AutoMQM predictions across all systems to perform meta-evaluation of this metric over the entire test set of interest. (Note that we also include a performance breakdown by system in §5.2.) See Tables 9 (WMT'23) and 10 (WMT'24) in Appendix B for the average number of ICL examples and average number of total errors across ICL examples per test example.

## 4.5. Meta-evaluation

To meta-evaluate Specialist AutoMQM, we compute the *character-level precision*, *recall*, and *F1* span tagging evaluation metrics (Blain et al., 2023). Given gold and predicted ratings, these metrics compute the precision, recall, and F1 of predicting whether a character in the hypothesis translation is included in an error span or not. Partial credit of 0.5 is given if the predicted rating correctly marks a character as an error but predicts the incorrect severity. To meta-evaluate the Specialist Scorer (which is not a span-based metric) in §5.6, we report segment-level pairwise accuracy with tie calibration (*Acc23*; Deutsch et al. (2023)), which was used to evaluate WMT'23 and WMT'24 Metrics Shared Task submissions. *Acc23* rewards metrics for correctly ranking translations and correctly predicting ties, in combination with a tie calibration procedure that introduces ties into metric scores so that the meta-evaluation is fairer.

## 5. Results and Discussion

The main results are shown in Table 1. First note that the "Shuffled sources" baseline (row 1c) already performs (at least) on par with the state-of-the-art AutoMQM models (XCOMET in row 1a and GEMBA in row 1b). Also note that the "Fixed, different source" baseline (row 1d) underperforms "Shuffled sources", which suggests that specializing to a different input is worse than no specialization. In contrast, the "Specialist" metric (row 2a) dramatically outperforms all of the baselines, with a 54% improvement in F1 score relative to XCOMET on WMT'23, and a 119% relative improvement on WMT'24. Recall that the difference between the "Specialist" setting and the "Fixed, different source" setting is that, in the former, ratings from translations of the *same source* as the test example are provided as demonstrations. Thus, same-source demonstrations are crucial to the success of our method, and its success cannot be attributed only to (i) providing demonstrations of ratings from different translations of some *fixed source*, or (ii) providing demonstrations from the *same rater* as the test rating ground truth.

## 5.1. Is Specialist AutoMQM Robust to the Choice of LLM?

The results reported in Table 1 used the Gemini 1.5 Pro LLM. To investigate whether gains from this method generalize to other LLMs, we also performed the same comparison using GPT-4o (Achiam et al., 2023) and Claude 3.5 Sonnet (Bai et al., 2022). As shown in Table 2, Specialist AutoMQM also substantially outperforms Shuffled AutoMQM when prompting these other LLMs. Moreover, note that Specialist AutoMQM with the GPT-4o backbone outperforms GEMBA (which is also a prompted GPT-4 model, albeit an earlier version) by an even larger margin. This supports the effectiveness of our approach over baselines using external (different-source) ICL examples. In the remaining experiments, we continue to use the Gemini 1.5 Pro LLM.

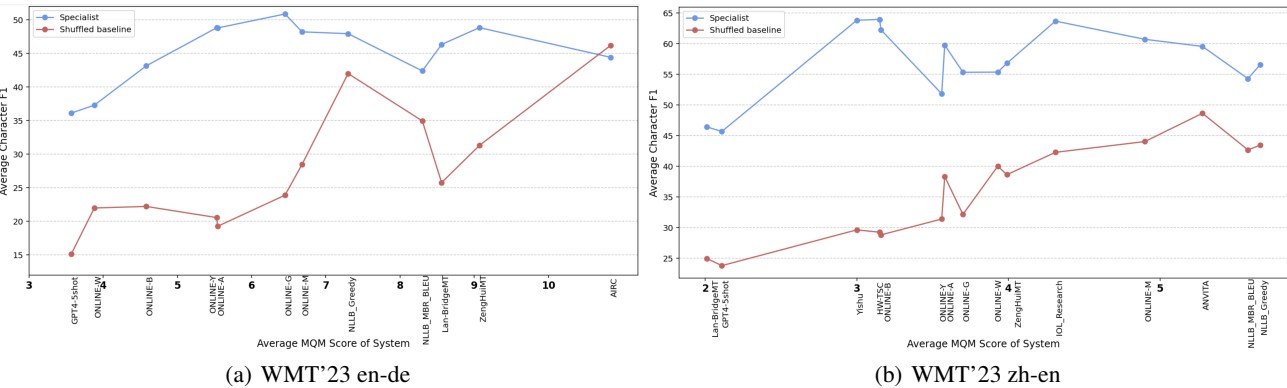

(a) WMT'23 en-de          (b) WMT'23 zh-en

*Figure 2.* Specialist AutoMQM performance per WMT'23 translation system. The Specialist and Shuffled baseline models from Table 1 (rows 2a and 1c, respectively) are compared. The average MQM score of each system, along with its name, is shown on the x-axis. The average character-level F1 of the AutoMQM model when evaluating this system only is shown on the y-axis.

| Character-level F1 | en→de | zh→en |
|---|---|---|
| **Baselines** | | |
| XCOMET-XXL-QE | 32.71 | 34.29 |
| GEMBA-MQM-QE | 29.80 | 34.17 |
| **Gemini-AutoMQM** | | |
| Shuffled sources | 31.49 | 37.80 |
| Specialist AutoMQM | 45.71 | **57.47** |
| **GPT-4o-AutoMQM** | | |
| Shuffled sources | 33.58 | 38.38 |
| Specialist AutoMQM | 48.32 | 56.78 |
| **Claude-3.5-Sonnet-AutoMQM** | | |
| Shuffled sources | 35.02 | 40.86 |
| Specialist AutoMQM | **48.49** | 56.13 |

*Table 2.* Comparison of Specialist AutoMQM vs the shuffled baseline (WMT'23 test set) for three different LLMs: Gemini 1.5 Pro, GPT-4o, and Claude-3.5-Sonnet. For all three LLMs, Specialist AutoMQM substantially outperforms the shuffled baseline.

## 5.2. How Does Specialist AutoMQM Performance Vary Across Translation Systems?

In practice, the Specialist AutoMQM metric would likely be used to evaluate the quality of new translation system(s), given ratings from historical systems. The WMT'23 and WMT'24 datasets contain rated translations from at least a dozen systems per language pair (Table 8). These systems are of varying quality, and aggregate meta-evaluation of AutoMQM (Table 1) could hide per-system differences in metric performance. Here, we compare performance of Specialist AutoMQM against the "Shuffled sources" baseline on a per-system basis for WMT'23. As shown in Figure 2, Specialist AutoMQM outperforms the shuffled baseline for every zh→en system, and for every en→de system except the lowest-quality one. Thus, Specialist AutoMQM outperformance is consistent for translation systems across the quality spectrum, and cannot be explained by gains only

for a certain quality tier. Note that both the Specialist and shuffled baseline AutoMQM models tend to perform worse on the highest-quality systems (e.g. GPT4-5shot), likely due to limitations in the underlying translation capabilities of the backbone language model used for AutoMQM.

## 5.3. How Does Specialist AutoMQM Performance Scale as a Function of Number of ICL Examples?

The most expensive and time-consuming step in the development of a Specialist metric is collecting ratings to use as demonstrations. Thus, it is useful to understand the marginal improvements in performance that can be expected as a result of collecting additional ratings. In this ablation, we randomly select subsets of Specialist AutoMQM ICL examples in the range [1, num_systems - 1]. While scaling ICL examples, we incrementally add a single new example to the existing set (for each test example), so that every set of ICL examples of a given size $n$ is a superset of the ICL examples for all sizes less than $n$. Note that when the number of ICL examples reaches its maximum (of num_systems - 1), this corresponds to the results reported in row 2a of Table 1. As shown in Figure 3, increasing the number of ICL examples improves character-level F1 monotonically up to 7 ICL examples for en→de, and up to 12 ICL examples for zh→en. We see that only 3 ICL examples are needed for Specialist AutoMQM to outperform the XCOMET baseline (for both en→de and zh→en) and that Specialist AutoMQM outperforms the "Shuffled sources" setting (Table 1, row 1c) at every ICL example set size.

## 5.4. Is Specialist AutoMQM Simply Copying Errors From ICL Examples?

### 5.4.1. LEARNING WHEN TO ABSTAIN

In view of the results from §5.3, i.e., that increasing the number of ICL examples improves performance of Spe-

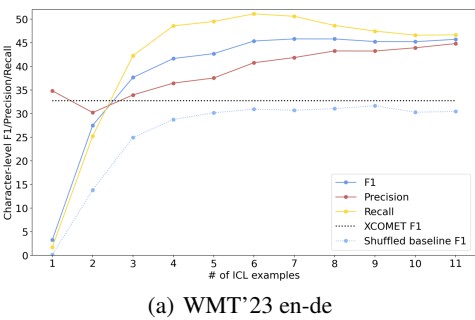
(a) WMT'23 en-de

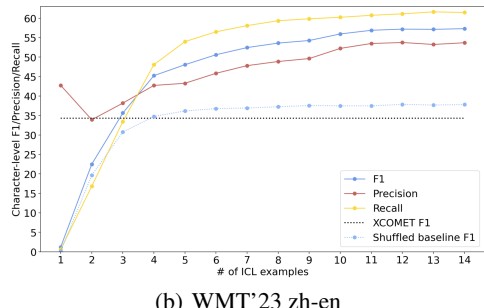
(b) WMT'23 zh-en

*Figure 3.* Specialist AutoMQM performance as a function of number of ICL examples used. For comparison, XCOMET performance, as well as ICL example scaling for the "Shuffled sources" baseline (Table 1, row 1c), are also shown.

cialist AutoMQM, this raises the question of whether the gains are simply due to the model copying errors that it is shown in the ICL examples. As shown in Table 14 in Appendix B (left two columns), when comparing Specialist AutoMQM prompted with 3 versus 11 ICL examples, the AutoMQM system prompted with 11 ICL examples predicts *fewer* errors which are direct copies of spans from all 11 ICL examples, than the AutoMQM system prompted with 3 ICL examples. The comparison is even more pronounced when comparing the "Shuffled Sources" baseline against Specialist AutoMQM with 11 ICL examples (right two columns of Table 14). After removing errors predicted by both systems, the shuffled baseline predicts 6,374 errors which are direct copies of spans from Specialist ICL examples, while Specialist AutoMQM only predicts 494 such errors (even though Specialist AutoMQM predicts more total errors than the shuffled baseline). Thus, Specialist AutoMQM is *abstaining* from predicting errors that it is shown via ICL examples, while the shuffled baseline, which has not been shown these errors, is predicting them more liberally.

### 5.4.2. PARROT MODEL BASELINE

To quantify how much of Specialist AutoMQM's performance can be attributed to error copying from ICL examples, we construct an artificial baseline, which we call the "Parrot". This model has access to the same ICL examples as Specialist AutoMQM, and makes predictions as follows: For every error present in ICL examples for which there is a matching span in the test translation, predict this as an error. As shown in Table 3, there is a large gap in character-level F1 between the Parrot model and Specialist AutoMQM (27.6 vs 45.7 for WMT'23 en→de, and 36.5 to 57.5 for WMT'23 zh→en). Thus, the performance of Specialist AutoMQM cannot be solely explained by naive copying behavior. The large gap in *recall* between the Parrot model and Specialist AutoMQM (see Table 15 in Appendix B) quantifies the extent to which Specialist AutoMQM correctly identifies errors not present in ICL examples. Examples of such AutoMQM predictions are shown in Table 16 (Appendix B).

The gap in *precision*, on the other hand, quantifies the extent to which Specialist AutoMQM correctly abstains from predicting errors present in ICL examples. See examples of such predictions in Table 17 (Appendix B).

| Character-level F1 | en→de | zh→en |
|---|---|---|
| Shuffled sources | 31.49 | 37.80 |
| Parrot | 27.59 | 36.52 |
| Specialist AutoMQM | 45.71 | 57.47 |

*Table 3.* "Parrot" vs Specialist AutoMQM performance (WMT'23).

### 5.5. Is Specialist AutoMQM Specialized Only to Test Sets, or Also to Raters?

By construction, Specialist AutoMQM is specialized to a test set. Since the WMT'23 and WMT'24 test sets are constructed in a pseudo-SxS fashion (with the exception of WMT'24 en→es), Specialist AutoMQM is also prompted with ICL examples rated by the same rater as the test rating ground truth. Here, we seek to understand whether Specialist AutoMQM also specializes to the rater. To answer this question, we take advantage of the additional rounds of WMT'23 MQM ratings, as described in §4.3.1.

| Character-level F1 | en→de | zh→en |
|---|---|---|
| **Human agreement** | | |
| 1a) Round2 | 34.91 | 38.68 |
| 1b) Round3 | 38.46 | 39.16 |
| **Specialist** | | |
| 2a) Round1 ICL | 45.71 | 57.47 |
| 2b) Round2 ICL | 30.80 | 38.83 |

*Table 4.* Specialist AutoMQM performance when prompting and evaluating using different raters (WMT'23 test set). Results are reported using the official Round1 test set.

**Prompting with Different Raters**   In the first set of experiments, we use the Specialist AutoMQM set-up, but prompt with ICL examples from Round2 (rather than Round1) ratings. We always evaluate using the official Round1 ratings.

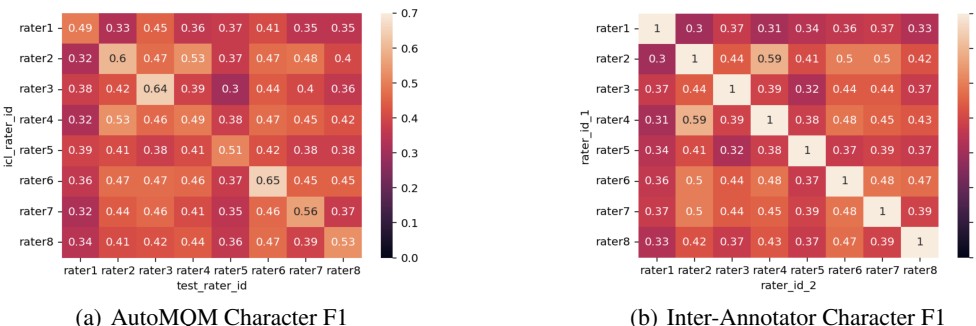

(a) AutoMQM Character F1        (b) Inter-Annotator Character F1

*Figure 4.* Cross-rater performance of AutoMQM and human annotators on the zh→en WMT'23 Multi-Rater Subset (§4.3.1). In Figure (a), Specialist AutoMQM is prompted with ICL examples from the icl_rater_id rater (vertical axis), and evaluated using the ratings from the test_rater_id rater (horizontal axis). In Figure (b), the matrix of character-level F1 scores between all pairs of human annotators is shown.

As shown in Table 4 (row 2b), performance drops to that of the "Shuffled sources" baseline (Table 1, row 1c). Thus, Specialist AutoMQM also specializes to the rater (on a per-example basis; recall that with the pseudo-SxS setup, raters can still vary across different inputs). This is not surprising, since there are large (and often competing) differences in behavior across raters: See rows 1a) and 1b) in Table 4 for the inter-annotator agreement across rounds. Note that the Round2 Specialist (row 2b) performs on par with the inter-annotator agreement for zh→en, and the Round1 Specialist (row 2a) outperforms the inter-annotator agreement for both language pairs, likely because it is able to match specific rater behavior from the ICL examples. In a follow-up experiment, we provide merged ratings from Round2 and Round3 as ICL examples. As shown in Table 18 in Appendix B, merging ratings (row 2c) improves recall at the cost of lower precision. The drop in precision likely represents not an actual quality drop, but under-annotation of errors (low recall) by the Round1 raters (used as the ground truth). It remains an open question how to combine ratings from multiple raters to create a better ground truth.

**ICL Rater × Test Set Rater Comparison** The aggregate results reported in Table 4 could mask individual cross-rater dynamics. While we do not have access to ratings from all raters for every system output across all test set examples, we do have access to the WMT'23 Multi-Rater Subset ratings (§4.3.1), for which all 8 raters rated 10% of the zh→en dataset. We use this WMT'23 Multi-Rater Subset to understand AutoMQM performance for *all pairs* of raters (where one rater is the ICL example annotator and the other rates the test set example), by computing the entire num_raters × num_raters matrix of F1 scores for every (ICL rater, test set rater) pair. These results are shown in Figure 4(a). As expected, the F1 scores on the matrix diagonal are highest (though note that rater2 and rater4 also have high agreement). For comparison against AutoMQM, Figure 4(b) shows the inter-annotator agreement (character-level F1) over the same WMT'23 Multi-Rater Subset. Observe

that when prompting and evaluating with different raters, on average Specialist AutoMQM agrees with the raters as much as the raters agree with each other.

| Segment-level Acc23 | WMT'23 | | WMT'24 | | |
| --- | --- | --- | --- | --- | --- |
| | en→de | zh→en | en→de | en→es | ja→zh |
| MetricX-24-QE | **59.44** | 54.48 | 52.45 | 68.48 | 52.69 |
| **AutoMQM** | | | | | |
| Shuffled sources | 53.55 | 49.62 | 50.39 | 68.21 | 51.57 |
| Specialist | 58.13 | **57.79** | **60.38** | **68.91** | 55.01 |
| **Score Prediction** | | | | | |
| Shuffled sources | 52.32 | 49.32 | 47.96 | 68.43 | 52.58 |
| Specialist | 56.77 | 55.93 | 56.78 | 68.58 | **56.56** |

*Table 5.* Comparison of Specialist AutoMQM vs the Specialist Scorer, on the WMT'23 and WMT'24 test sets.

### 5.6. Does the Specialist Method Generalize to Other Automatic Evaluation Tasks?

**Translation Direct Assessment** We have seen that the Specialist method for prompting LLMs-as-Judges achieves state-of-the-art performance for the task of AutoMQM. Here, we consider the task of scoring translation quality (without providing error annotations). In this task, we prompt the LLM to generate a float quality score on a scale from 0-100. (See Figure 6 in Appendix A for the prompt used.) As shown in Table 5, the task of score prediction also benefits substantially from the Specialist method (relative to the shuffled baseline). Here, we report segment-level accuracy, and also compare against MetricX-24 (Juraska et al., 2024), the state-of-the-art automatic score prediction metric for machine translation. Also note that Specialist AutoMQM has a quality advantage over the Specialist Scorer, while also offering the added benefit of interpretability (perhaps because LLMs are better at natural language text generation versus generation of numbers). Finally, observe that

|  | Relevance (excl. HS) | | Relevance (incl. HS) | | Average (excl. HS) | | Average (incl. HS) | |
|---|---|---|---|---|---|---|---|---|
|  | Pearson | Spearman | Pearson | Spearman | Pearson | Spearman | Pearson | Spearman |
| Shuffled baseline | 43.71 | 41.51 | 59.58 | 52.88 | 36.74 | 32.35 | 54.32 | 44.40 |
| Specialist | 49.01 | 46.99 | 63.76 | 57.90 | 46.34 | 42.36 | 66.16 | 54.78 |

*Table 6.* Story-level Pearson and Spearman correlations of the Shuffled baseline vs Specialist metric, evaluated on the HANNA benchmark. We report results for both the relevance criterion and the average over all criteria, both with human stories included (indicated in table by *incl. HS*) and without human stories (*excl. HS*).

both Specialist models outperform MetricX-24 across all WMT'23 and WMT'24 language pairs except WMT'23 en→de.

**Long-Form Story Generation Assessment** Both tasks considered so far have evaluated translation quality (via either error annotation or direct assessment). To validate the generalization of the Specialist method, here we consider the task of story generation, which is an open-ended NLG task with fundamentally different characteristics from translation evaluation. To evaluate story generation, we use the HANNA benchmark (Chhun et al., 2022), which satisfies the requirements of the Specialist method since it contains outputs from multiple systems for every input prompt, augmented with human annotations. While the annotations were not collected in pseudo-SxS fashion (§4.3), each output was annotated by three human raters, and (following Chhun et al., 2022) we average over these ratings during meta-evaluation to smooth out inter-rater variability. The annotations evaluate each story according to six criteria: relevance, coherence, empathy, surprise, engagement, and complexity. We evaluate our baseline and Specialist metrics on the first criterion (relevance) and on the average score over all criteria (which represents an overall indication of the story's quality), and report story-level Pearson and Spearman correlations.

We follow the same experimental methodology as described in Section 4.4, and construct Specialist ICL examples for each test example via hold-one-system-out prompting: that is, for whichever system output we are evaluating, we exclude that system's ratings from the ICL examples and prompt with the ratings from the remaining systems. This simulates the real-world use case of evaluating a new system. Moreover, as described in Section 4.2, we enforce the following constraint when constructing ICL examples for the Shuffled baseline: The ICL examples for a given test example cannot include any outputs, whether from the same or different prompt, produced by the same system as that which produced the test output. We evaluate both with and without human stories included, since the HANNA paper (Chhun et al., 2022) excluded human stories in its meta-evaluation (as these were considered outliers), while some

follow-up papers, such as Yuan et al. (2023), included them.

As shown in Table 6, the Specialist metric beats the Shuffled baseline according to both criteria (relevance and average score) and both meta-evaluation metrics (story-level Pearson and Spearman correlations). For average score prediction (excluding human stories), the Specialist metric outperforms the Shuffled baseline by 26.1% according to Pearson correlation, and by 30.9% according to Spearman correlation.

# 6. Conclusion

In this work, we have proposed the *Specialist* method for development of automatic evaluation metrics which are specialized to a given test set. We have shown that Specialist AutoMQM dramatically outperforms all existing state-of-the-art span-based MT evaluation metrics, on both the WMT'23 and WMT'24 test sets. Specialist evaluators are easy to implement, as they are simply multi-shot prompted LLMs. Moreover, the Specialist method is task-agnostic, and an immediate avenue for future work would be to apply this method to development of metrics for other NLG tasks. These Specialist metrics could serve as a powerful alternative to human judges in evaluating LLM quality across a wide range of capabilities. Another avenue for future work is to better understand how to combine ratings from multiple raters, both for creation of ICL examples for Specialist metrics, and for creation of more trustworthy test sets (which are capable of measuring super-human performance). Finally, the Specialist method as framed here requires human-generated ratings to be used as ICL examples, but future work could explore whether LLMs are also capable of generating these ratings.

## Impact Statement

This paper presents work whose goal is to advance the field of Machine Learning. There are many potential societal consequences of our work, none which we feel must be specifically highlighted here.

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

## A. Implementation Details

Figure 5 shows the AutoMQM prompt template, Table 7 shows an example AutoMQM output, and Figure 6 shows the direct assessment scoring prompt template.

---

**AutoMQM Prompt Template**

```
You are an annotator for the quality of machine translation. Your task is to
identify errors and assess the quality of the translation.
Based on the source segment and machine translation surrounded with triple
backticks, identify error types in the translation and classify them. The
categories of errors are: accuracy (addition, mistranslation, omission,
untranslated text), fluency (character encoding, grammar, inconsistency,
punctuation, register, spelling), style (awkward), terminology (inappropriate
for context, inconsistent use), non-translation, other, or no-error.
Each error is classified as one of three severities: critical, major, and
minor. Critical errors inhibit comprehension of the text. Major errors disrupt
the flow, but what the text is trying to say is still understandable. Minor
errors are technically errors, but do not disrupt the flow or hinder
comprehension.

Make sure your response is a strict and valid json object that could be parsed
with json.loads() in python.
```

**ICL examples**

```
{source_language} source:
```{source}```
{target_language} translation:
```{translation}```
{errors in JSON format}

{source_language} source:
```{source}```
{target_language} translation:
```{translation}```
{errors in JSON format}
```

**Test example**

```
{source_language} source:
```{source}```
{target_language} translation:
```{translation}```
```

---

*Figure 5.* AutoMQM prompt, with placeholders for {source_language}, {source} (for both ICL examples and the test example), {target_language}, {translation} (again, for both ICL examples and the test example), and {errors in JSON format} (for ICL examples only).

| Source | 害得我，从外地驱车200公里赶回来取货！ |
|---|---|
| Hypothesis | I'm sorry that we had to drive 200 kilometers from the country to pick up my goods! |
| Output | [{"span": "I'm sorry that", "severity": "minor", "category": "style/unnatural or awkward"}, |
| | {"span": "we", "severity": "minor", "category": "accuracy/mistranslation"}, |
| | {"span": "the country", "severity": "major", "category": "accuracy/mistranslation"}] |

*Table 7.* Example Specialist AutoMQM output (from the WMT'23 zh→en test set). As per the AutoMQM prompt (Figure 5 in Appendix A), the output is in JSON format, with fields for error span, severity, and category. The highlighting is added to the hypothesis for illustrative purposes, to indicate the locations of the predicted major (dark red) and minor (light red) errors.

---

**Direct Assessment Scoring Prompt Template**

```
You are a judge for the quality of machine translation. Based on the
source segment and machine translation surrounded with triple backticks,
your task is to assess the quality of the machine translation on a
continuous scale from 0 to 100. A score of 0 means "No meaning preserved",
then the scale goes through "Some meaning preserved", to "Most meaning
preserved and few grammar mistakes", up to a score of 100, which means
"Perfect meaning and grammar".
```

**ICL examples**

```
{source_language} source:
'''{source}'''
{target_language} translation:
'''{translation}'''
Score: [[{score}]]

{source_language} source:
'''{source}'''
{target_language} translation:
'''{translation}'''
Score: [[{score}]]
```

**Test example**

```
{source_language} source:
'''{source}'''
{target_language} translation:
'''{translation}'''
```

*Figure 6.* Direct Assessment prompt, with placeholders for {source_language}, {source} (for both ICL examples and the test example), {target_language}, {translation} (again, for both ICL examples and the test example), and {score} (for ICL examples only).

# B. Supplemental Results

## B.1. Additional Figures and Tables

### B.1.1. EXPERIMENTAL SETUP

| Number of systems | WMT'23 | WMT'24 |
|---|---|---|
| en→de | 12 | 17 |
| zh→en | 15 | N/A |
| en→es | N/A | 13 |
| ja→zh | N/A | 13 |

Table 8. Number of translation systems for each language pair (WMT'23 and WMT'24)

| Avg # ICL examples / Avg # errors per test example | en→de | zh→en |
|---|---|---|
| No filtering | 11.0/34.8 | 14.0/30.7 |
| Filtered | 10.5/27.4 | 13.3/25.8 |

Table 9. Average number of ICL examples and average number of total errors in ICL examples, per test example (WMT'23 test set). The filtered setting removes all translations from ICL examples which are exact matches to the test translation, and removes all individual errors from ICL examples which exactly match a ground-truth error span in the test translation.

| Avg # ICL examples / Avg # errors per test example | en→de | en→es | ja→zh |
|---|---|---|---|
| No filtering | 17.0/22.8 | 13.0/5.8 | 13.1/13.3 |
| Filtered | 16.0/19.4 | 12.4/5.6 | 12.8/12.8 |

Table 10. Average number of ICL examples and average number of total errors in ICL examples, per test example (WMT'24 test set). The filtered setting removes all translations from ICL examples which are exact matches to the test translation, and removes all individual errors from ICL examples which exactly match a ground-truth error span in the test translation.

### B.1.2. MAIN RESULTS

| | en→de | | | zh→en | | |
|---|---|---|---|---|---|---|
| | F1 | Precision | Recall | F1 | Precision | Recall |
| **Baselines** | | | | | | |
| 1a) XCOMET-XXL-QE | 32.71 | 28.66 | 38.10 | 34.29 | 39.70 | 30.18 |
| 1b) GEMBA-MQM-QE | 29.80 | 32.04 | 27.85 | 34.17 | 39.87 | 29.89 |
| 1c) Shuffled sources | 31.49 | 28.34 | 35.45 | 37.80* | 32.79* | 44.62* |
| 1d) Fixed, different source | 22.85 | 26.08 | 20.35 | 31.27 | 34.54 | 28.58 |
| **Specialist** | | | | | | |
| 2a) Specialist | **45.71*** | **45.04*** | **46.40*** | **57.47*** | **54.05*** | **61.36*** |
| 2b) Specialist + Filter | 38.32* | 39.05* | 37.61* | 50.72* | 49.32* | 52.21* |

Table 11. Specialist AutoMQM results on the WMT'23 test set. See §4.2 for a description of all of the Baseline and Specialist systems, and see §B.2.1 for a description of the filtered setting. Results for the "Shuffled sources" and "Fixed, different source" baselines are reported as the average over 10 runs with different random seeds. See Table 12 for the variance across runs. Asterisks (∗) indicate scores which are statistically significantly better than XCOMET (row 1a), according to a paired permutation test.

| | | en→de | | | zh→en | | |
|---|---|---|---|---|---|---|---|
| | | F1 | Precision | Recall | F1 | Precision | Recall |
| Shuffled sources | AVG | 31.49 | 28.34 | 35.45 | 37.80 | 32.79 | 44.62 |
| Shuffled sources | STDEV | 0.71 | 0.62 | 1.01 | 0.33 | 0.33 | 0.50 |
| Fixed, different source | AVG | 22.85 | 26.08 | 20.35 | 31.27 | 34.54 | 28.58 |
| Fixed, different source | STDEV | 0.76 | 0.93 | 0.97 | 0.48 | 0.57 | 0.67 |

*Table 12.* Average (AVG) character-level F1, precision, and recall over 10 runs of the "Shuffled sources" and "Fixed, different source" baselines with different random seeds. Standard deviation (STDEV) over the 10 runs is also reported.

| | en→de | | | en→es | | | ja→zh | | |
|---|---|---|---|---|---|---|---|---|---|
| | F1 | Precision | Recall | F1 | Precision | Recall | F1 | Precision | Recall |
| **Baselines** | | | | | | | | | |
| 1a) XCOMET-XXL-QE | 24.28 | 19.63 | 31.83 | 10.11 | 6.02 | 31.42 | 14.30 | 11.80 | 18.16 |
| 1b) Shuffled sources | 26.12 | 19.67 | 38.84 | 26.12* | 19.67* | 38.84* | 26.44* | 32.46* | 22.30* |
| 1c) Fixed, different source | 18.23 | 19.82 | 16.87 | 14.89 | 11.09 | 22.65 | 26.77* | 32.25* | 22.89* |
| **Specialist** | | | | | | | | | |
| 2a) Specialist | **43.04*** | **39.16*** | **47.76*** | **26.58*** | **20.05*** | **39.43*** | **37.16*** | **38.06*** | **36.30*** |
| 2b) Specialist + Filter | 32.83* | 31.07* | 34.79* | 25.58* | 19.34* | 37.79* | 35.73* | 36.83* | 34.69* |

*Table 13.* Specialist AutoMQM results on the WMT'24 test set. Note that en→es ratings were *not* collected in a *pseudo-SxS* fashion (see §4.3), which explains the smaller performance delta between the Specialist method and the baselines for this language pair. See §4.2 for a description of all of the Baseline and Specialist systems, and see §B.2.1 for a description of the filtered setting. Asterisks (∗) indicate scores which are statistically significantly better than XCOMET (row 1a), according to a paired permutation test. Note: The en→es MQM data was not collected in a pseudo-SxS fashion, so ratings from different raters were presented as ICL examples in the Specialist setup for this language pair.

### B.1.3. ERROR COPYING ABLATIONS

| | 3 ICL Examples | 11 ICL Examples | Shuffled | 11 ICL Examples |
|---|---|---|---|---|
| 1) Total predicted error count | 14,539 | 14,481 | 10,298 | 14,481 |
| 2) Disjoint error count | 9,185 | 9,127 | 7,913 | 12,096 |
| 3) Disjoint error count with exact match to ICL example errors | 4,142 | 2,327 | 6,374 | 494 |

*Table 14.* Pairwise comparison of predicted errors copied from ICL examples, for different AutoMQM systems. The left two columns show a comparison of the Specialist AutoMQM (Table 1, row 2a) prompted with 3 vs 11 ICL examples, and the right two columns show a comparison of the latter against the shuffled baseline (Table 1, row 1c). Row 1 shows the total predicted error count over the full WMT'23 en→de test set, row 2 shows the number of errors predicted by the given system which were not predicted by the other system being compared, and row 3 shows the subset of these errors which are exact matches to errors from all 11 (same-source) ICL examples.

| | en→de | | | zh→en | | |
|---|---|---|---|---|---|---|
| | F1 | Precision | Recall | F1 | Precision | Recall |
| Shuffled sources | 31.49 | 28.34 | 35.45 | 37.80 | 32.79 | 44.62 |
| Parrot | 27.59 | 25.62 | 29.88 | 36.52 | 34.92 | 38.27 |
| Specialist AutoMQM | 45.71 | 45.04 | 46.40 | 57.47 | 54.05 | 61.36 |

*Table 15.* "Parrot model" vs Specialist AutoMQM performance (WMT'23 test set). The Parrot has access to the same ICL examples as Specialist AutoMQM, and makes predictions as follows: For every error present in ICL examples for which there is a matching span in the test translation, predict this as an error.

| | |
|---|---|
| Source | 34CM的床垫不是一般的厚，不要床直接睡床垫都可以了。 |
| Test Example Hypothesis | A 34CM mattress `is not usually thick`, `so it is not necessary to place the bed directly on the mattress`. |
| ICL Examples | A 34cm mattress is not `typically` thick, `you` could even sleep directly on the mattress without a bed. |
| | `The 34CM mattress is not generally thick`, and you can sleep directly on the mattress without a bed. |
| | The 34CM mattress is unusually thick, `you` can sleep directly on the mattress without a bed. |
| | `34CM mattress is not generally thick, do not sleep directly on the mattress can be.` |
| | The 34cm mattress `is not usually thick`, so you can sleep directly on the mattress without the bed. |
| | `34 cm mattress is not as thick as usual`, but the beds can be used directly. |
| | The 34CM mattress `is not so thick`, you can just sleep on the mattress without the bed. |
| | . . . |
| Source | 吓得我把收藏夹里的其乐都删了。 |
| Test Example Hypothesis ICL Example Errors | I was so scared that I deleted all the `games` from my favorites. |
| | I was so scared that I deleted all the `music` in my favorites. |
| | I was so scared that I deleted all the `fun` in my favorites. |
| | I'm `afraid` I deleted all the `music` from my collection. |
| | `I was in the middle of a conversation.` |
| | I am so scared `to` remove all the `items` in my collection. |
| | `Scared` me so `much I` deleted all my favorites of `its music`. |
| | . . . |

*Table 16.* Examples of where Specialist AutoMQM predicts errors not present in ICL examples (WMT'23 zh→en test set). Green highlighting in the Test Example Hypothesis shows where Specialist AutoMQM correctly predicted an error span that was *not* present in the ICL examples, while red highlighting indicates a span (correctly) copied from ICL examples. Red highlighting in the ICL Examples indicates the error spans that were marked by human MQM annotators (and provided to Specialist AutoMQM as demonstrations).

| | |
|---|---|
| Source | 标题上还是顾客,正文中就变成客户了。 |
| ICL Example Hypothesis | The title is still a `customer`, but the text becomes a customer. |
| Test Example Hypothesis | The title still refers to the `customer`, but in the body of the text, it has changed to client. |
| Source | 让一个身上3处伤口的老人下床开门收快递还要找零钱付费！ |
| ICL Example Hypothesis | Let an old man with 3 wounds get out of bed and open the door to receive the courier and `change to pay`! |
| Test Example Hypothesis | To get an old man with 3 wounds on his body to get out of bed and open the door to receive the package, he still has to find `change to pay`! |

*Table 17.* Examples of where Specialist AutoMQM correctly abstains from copying errors in ICL examples (WMT'23 zh→en test set). Red highlighting indicates that the span was marked as an error by the human MQM annotators, and green highlighting indicates that the span was not marked as an error.

B.1.4. RATER ABLATIONS

| | en→de | | | zh→en | | |
|---|---|---|---|---|---|---|
| | F1 | Precision | Recall | F1 | Precision | Recall |
| **Human agreement** | | | | | | |
| 1a) Round2 | 34.91 | 38.16 | 32.17 | 38.68 | 39.00 | 38.36 |
| 1b) Round3 | 38.46 | 40.26 | 36.82 | 39.16 | 40.06 | 38.29 |
| **Specialist** | | | | | | |
| 2a) Round1 ICL | 45.71 | 45.04 | 46.40 | 57.47 | 54.05 | 61.36 |
| 2b) Round2 ICL | 30.80 | 30.87 | 30.74 | 38.83 | 36.65 | 41.30 |
| 2c) Round2 \| Round3 ICL | 30.83 | 26.65 | 36.56 | 38.48 | 29.91 | 53.93 |

*Table 18.* Specialist AutoMQM performance when prompting and evaluating using different raters (WMT'23 test set). "Round 2 | Round3" indicates that ratings from these rounds were merged. Results are reported using the official Round1 test set.

| | Round1 ICL Examples | | Round2 ICL examples | | |
|---|---|---|---|---|---|
| test_set_rater_id | icl_rater_id | F1 | icl_rater_id | F1 | num_examples |
| rater1 | rater1 | 0.39 | rater8 | 0.30 | 672 |
| rater2 | rater2 | 0.50 | rater6 | 0.35 | 540 |
| rater3 | rater3 | 0.45 | rater1 | 0.33 | 564 |
| rater4 | rater4 | 0.46 | rater10 | 0.33 | 552 |
| rater5 | rater5 | 0.46 | rater7 | 0.26 | 588 |
| rater6 | rater6 | 0.51 | rater4 | 0.36 | 540 |
| rater7 | rater7 | 0.46 | rater9 | 0.29 | 492 |
| rater8 | rater8 | 0.37 | rater2 | 0.27 | 528 |
| rater9 | rater9 | 0.48 | rater3 | 0.34 | 528 |
| rater10 | rater10 | 0.51 | rater5 | 0.26 | 516 |

*Table 19.* Specialist AutoMQM performance on WMT'23 en→de Round1, when prompting using Round1 vs Round2 ICL examples, broken out by rater split. In each round of WMT'23 ratings, there are a total of 10 en→de raters. The examples in the test set are then approximately split evenly across all raters (such that all translations of the same source segment are allocated to the same rater). Note that the variance across raters when using different-rater (Round2) ICL examples is not very high, and using Round1 ICL examples outperforms Round2 ICL examples for every split

## B.2. Additional Ablations

B.2.1. FILTERING ICL EXAMPLES TO REMOVE EXACT-MATCH ERRORS

Here, we isolate the effect on performance of showing Specialist AutoMQM errors in ICL examples which are an exact match to a ground-truth error in the test translation. In particular, we filter ICL examples to i) remove errors with the same span (but not necessarily the same category or severity) as ground-truth errors present in the test translation, and ii) entirely exclude all translations (rather than just removing exact-match errors) in ICL examples which are exact matches to the test translation.

As expected, filtering the ICL examples by removing all error spans present in the ground truth ("Specialist + Filter" setting, row 2b in Tables 11 and 13 for WMT'23 and WMT'24, respectively) does incur some degradation in performance relative to the "Specialist", but still significantly outperforms all baselines, including the state-of-the-art XCOMET and "Shuffled source" models. Also note that filtering to remove individual errors from ICL examples in some sense unfairly disadvantages the model, since this procedure excludes real errors from the demonstrations, and these are, in fact, exactly those errors which would be correct for the model to predict.

For this filtered Specialist AutoMQM, we computed exact match rates with respect to i) ground truth errors spans in the test translations and ii) error spans present in ICL examples. The results are shown in Table 20. Observe that, even though the model was not shown any of the ground truth errors in the provided demonstrations, 17.1% (for en→de) and 23.7% (for

| Exact Match Error % | en→de | zh→en |
|---|---|---|
| 1) Ground Truth | 17.10 | 22.70 |
| 2) ICL Examples | 27.52 | 26.10 |
| 3) ICL Examples (incl. sub-span + super-span) | 65.25 | 68.79 |

*Table 20.* Exact match error rate of Specialist AutoMQM predictions, as a percentage of total predicted errors, with respect to the ground truth error spans (row 1) and error spans present in ICL examples (row 2). Row 3 shows the match rate when predicted error spans which are either sub-spans or super-spans of errors present in ICL examples are also counted as matches. Results are presented for "Specialist + Filter" WMT'23 en→de Specialist AutoMQM (Table 13, row 2b), so the model is not shown demonstrations of any errors with exact match to the ground truth errors in the test translation.

zh→en) of the errors that it predicts are exact matches to the ground truth, while 26-27% of the errors that it predicts are exact matches to ICL example errors spans. If predicted error spans which are either sub-spans or super-spans of errors present in ICL examples are also counted as matches, then the match rate more than doubles, to 65-68%. This suggests that Specialist AutoMQM is also taking into account the semantics of the errors in the ICL examples, and is able to generalize its predictions to account for modified versions of these errors present in the test translations.

### B.2.2. CAN WE DO BETTER? AUGMENTING SPECIALIST AUTOMQM WITH MORE ICL EXAMPLES

Specialist AutoMQM only uses the same-source ratings from the test set as ICL examples, which limits the number of ICL examples to num_systems - 1. Modern LLMs can handle much longer context than these examples occupy, so in this ablation, we investigate whether augmenting the same-source ICL examples provided to Specialist AutoMQM with other ICL examples from the test set can further enhance performance. In particular, for each test set example, we first provide the ICL examples from the shuffled baseline, then concatenate the ICL examples from Specialist AutoMQM. As shown in Table 21, augmenting Specialist AutoMQM with additional examples results in a small *drop* in character-level F1, due to lower recall (despite a small improvement in precision). Recall that in Figure 3, we also saw that Specialized AutoMQM performance saturates at around 10 (same-source) ICL examples. This suggests that there is not substantial headroom to improve AutoMQM's performance by filling up the LLM's long context window, either with same-source or difference-source ICL examples.

| | en→de | | | zh→en | | |
|---|---|---|---|---|---|---|
| | F1 | Precision | Recall | F1 | Precision | Recall |
| Shuffled sources | 31.12 | 27.93 | 35.13 | 37.62 | 32.45 | 44.74 |
| Specialist AutoMQM | **45.71** | 45.04 | **46.40** | **57.47** | 54.05 | **61.36** |
| Shuffled Sources + Specialist AutoMQM | 44.75 | **46.17** | 43.42 | 57.36 | **55.79** | 59.02 |

*Table 21.* Comparison of prompting with i) only shuffled sources, ii) only same-source examples, or iii) both. Adding additional ICL examples gives higher precision at the cost of lower recall.

