# OpenReview forum: "From Jack of All Trades to Master of One: Specializing LLM-based Autoraters to a Test Set"
_ICML.cc/2025/Conference — ICML 2025 poster_

### Official Review · Reviewer_19zf · 2025-03-08

**Overall Recommendation:** 3

**Summary:**

While traditional LLM evaluators (Autoraters) are generally trained for generalization to a broad set of tasks, this paper studies specializing existing models into autoraters for particular known test sets. In particular, the proposed new prompting strategy leverages in-context learning examples obtained from historical ratings of each individual test set example. Empirically, the authors focus on the machine translation task, and show substantial improvements meta-evaluating their metric against past considered autorater methods.

**Claims And Evidence:**

The claims are supported by solid empirical evidence (see Experiments Section below).

**Essential References Not Discussed:**

I believe the paper discusses most of the relevant prior work.

**Experimental Designs Or Analyses:**

The experimental breadth is sound. The authors apply their method prompting a powerful Gemini LLM as the autorater. They evaluate their method against relevant recent baselines across 5 different target/source language combinations.

On top of convincing results, the authors also provide interesting parameter studies to highlight the sensitivities of their method (e.g., performance as a function of the base LLM autorater/number of in-context samples) and evaluate potential failure modes (e.g, compare performance vs. 'Parrot' baseline also copying the in context example's errors).

**Methods And Evaluation Criteria:**

The proposed improvement is simple and very intuitive.

I find the practical value of the proposed method is hindered by its inherent requirements and costs of having full access to the evaluation set of examples. I think this would make it inappropriate to rate specialized models and applications where successful translations can be hard to obtain or costly (e.g., underrepresented languages, long passages).

The scalability is also a potential concern, as with more comprehensive and larger test sets, there will be linearly scaling costs.

**Other Comments Or Suggestions:**

No additional comments.

**Other Strengths And Weaknesses:**

The paper is clear and well-written.

The novelty of the methodology is relatively limited due to the incremental nature of the work.

**Questions For Authors:**

No additional questions.

**Relation To Broader Scientific Literature:**

The work is a direct extension of prior prompt-based LLM autoraters systems for machine translation. The main difference with prior work such as [1, 2] is the use of in-context examples that are specific to each individual sample in the test set.

[1] Fernandes, Patrick, et al. "The devil is in the errors: Leveraging large language models for fine-grained machine translation evaluation." arXiv preprint arXiv:2308.07286 (2023)

[2] Kocmi, Tom, and Christian Federmann. "GEMBA-MQM: Detecting translation quality error spans with GPT-4." arXiv preprint arXiv:2310.13988 (2023).

**Theoretical Claims:**

The paper does not make theoretical claims.

---

> ### Author Rebuttal · Authors · 2025-03-31
>
> Dear Reviewer, thank you very much for your careful and comprehensive review of our paper. We appreciate that you recognize our sound experimental setup and clear presentation, as well as convincing results and parameter studies. We would like to address your concerns about the practicality of the Specialist method.
> > I find the practical value of the proposed method is hindered by its inherent requirements and costs of having full access to the evaluation set of examples.
>
> It is true that the Specialist method specializes to a fixed test set by design. By removing the constraint that the metric generalize to new test sets, we achieve large gains in the metric's ability to generalize to new systems, and this is often a tradeoff that is worth making. It is common to hillclimb for many years on a fixed test set (e.g., WMT for machine translation, XLSum for Summarization, SQuAD for question answering). Eventually these benchmarks saturate and must be refreshed, but benchmarks for core NLG tasks are often carefully curated to measure certain capabilities of interest and are repeatedly used to allow for fair comparison against previous work (including during the model development process). Thus, the Specialist method provides a paradigm for joint development of Automatic Metrics and Test Sets (which are typically developed independently, but don't need to be, and we get quality gains if they're developed jointly).
>
> > I think this would make it inappropriate to rate specialized models and applications where successful translations can be hard to obtain or costly (e.g., underrepresented languages, long passages).
>
> We interpret this concern to relate to the cost of collecting outputs from multiple models on the given test set, for the sake of then collecting human annotations to construct Specialist ICL examples, but please correct if this is a misinterpretation. If "successful translations" are hard to obtain, this would apply equally to the collection of outputs for the creation of ICL examples as it would at test time. Thus, this begs the question of whether this task should be used for evaluation in the first place, or would be a bottleneck at every evaluation.
>
> The above point aside, it is true that for some NLG tasks, there are not that many different models/systems which can perform the given task (and from which to collect ICL examples for the Specialist Autorater). However, as we show in the ICL example scaling experiments in Section 5.3, only 3 ICL examples (i.e., ratings of outputs from 3 other systems) are needed to outperform the state-of-the-art. For many NLG tasks, including those tracked most closely on LLM leaderboards, we do indeed have easy access to outputs from multiple systems.
>
> Moreover, as discussed in the "Specialist Algorithm" paragraph in Section 3, while "the different translations for each input example come from different translation systems [in this work], they could in principle also be sampled from a single model (e.g., using a diversity promoting sampling algorithm)." This is a promising direction for future work, especially for tasks which are not well represented in available models.
>
> > The scalability is also a potential concern, as with more comprehensive and larger test sets, there will be linearly scaling costs.
>
> This work explores how to use human annotations to enhance the quality of automatic metrics and, while the cost of collecting Specialist annotations for a test set does scale linearly with the size, evaluation using automatic metrics is much cheaper than directly depending on human annotations for evaluation, without sacrificing quality (as shown in the comparison against inter-annotator agreement in Section 5.5).
>
> Moreover, with larger test sets, the imperative to depend on automatic metrics (rather than human evaluation) increases too. One-off collection of human ratings on a large(r) test set to build an automatic metric is much more tractable than repeated human evaluations on this test set. Our hope is that the results in our paper showing the effectiveness of the Specialist method will provide motivation to the broader NLG community to acquire human annotations and build Specialist metrics for other tasks and datasets.
>
> We hope our responses and clarifications have allayed your concerns about our method's practicality, and that you will consider increasing our score. Please let us know if you have any further questions or if we can provide any additional clarifications to help finalize your assessment of our paper.

---

> > ### Comment · Reviewer_19zf · 2025-04-05
> >
> > I am not entirely convinced by authors' arguments, but I understand their points. I think the paper provides value, as detailed in my review. Thus, I will keep my positive rating.

---

### Official Review · Reviewer_iJij · 2025-03-08

**Overall Recommendation:** 1

**Summary:**

This paper presents a novel approach to enhancing automatic evaluation metrics based on LLMs, focusing on Machine Translation (MT) evaluation. The authors propose a Specialist method that leverages historical human-generated ratings on a fixed test set to construct ICL demonstrations. This specialization allows an LLM-based Autorater, termed "Specialist AutoMQM," to outperform existing state-of-the-art metrics on fine-grained MT evaluation tasks.

**Claims And Evidence:**

The authors' main claim includes a framework for effectively conducting automatic evaluations of NLG tasks on a fixed set that remains largely unchanged. However, it seems this work includes several limitations.

The authors broadly claim the generalizability of the Specialist method across various NLG tasks. However, explicit experiments were limited to two closely related MT tasks (AutoMQM and Direct Assessment scoring). The method’s effectiveness on tasks with fundamentally different characteristics (e.g., open-ended generation, dialogue evaluation, summarization quality assessment) remains uncertain. Without explicit validation, these claims are somewhat overstated.

**Essential References Not Discussed:**

I think

**Experimental Designs Or Analyses:**

The experimental design is sound, involving rigorous ablations (e.g., varying ICL examples, "Shuffled sources," and "Fixed, different source" baselines), ensuring thorough validation of the effectiveness of same-source ICL specialization.

However, all experiments rely on WMT test sets, which, despite being standard and widely used, may not adequately reflect real-world translation scenarios or fully capture the variability found in practical applications. The performance improvements reported might therefore not fully generalize outside WMT benchmarks.

**Methods And Evaluation Criteria:**

The proposed Specialist method relies heavily on the availability of high-quality historical ratings from human annotators for each test set. This requirement is a major practical limitation in contexts where obtaining such annotations might be prohibitively expensive, labor-intensive, or challenging (e.g., low-resource languages or less common test sets).

In particular, the proposed method lacks any significant technical novelty. The approach of carefully adjusting ICL demonstrations to enhance or analyze the performance of specific tasks has already been extensively studied.

**Other Comments Or Suggestions:**

N/A

**Other Strengths And Weaknesses:**

The Specialist method depends on prompting large models like Gemini 1.5 Pro, GPT-4o, or Claude. In practice, large-scale deployment of prompted LLM-based metrics can be computationally costly. Moreover, evaluation latency, cost implications, and practicality for real-time evaluations during rapid model iterations have not been discussed.

**Questions For Authors:**

N/A

**Relation To Broader Scientific Literature:**

I could not find the any broader impact for this paper.

**Theoretical Claims:**

N/A

---

> ### Author Rebuttal · Authors · 2025-03-31
>
> Dear Reviewer, thank you very much for your feedback, which has helped us to improve our paper. In response to your primary concern, namely that the Specialist method's "effectiveness on tasks with fundamentally different characteristics (e.g., open-ended generation, [...]) remains uncertain", we have added a new experiment showing the effectiveness of the Specialist method on the HANNA benchmark (https://arxiv.org/pdf/2208.11646), which evaluates story generation. For additional details of our experimental setup and results, please see our response to Reviewer inJi. As shown in the table below (which reports story-level Pearson and Spearman correlations), our Specialist method demonstrated substantial gains on HANNA, outperforming the shuffled baseline across both tasks and all meta-evaluation metrics.
> |Metric|REL (excl. human)||REL (incl. human)| |AVG (excl. human)| |AVG (incl. human)| |
> |:-----------|:-------:|:-------:|:-------:|:-------:|:-------:|:-------:|:-------:|:-------:|
> ||r_p|r_s|r_p|r_s|r_p|r_s|r_p|r_s|
> |Shuffled baseline|43.71|41.51|59.58|52.88|36.74|32.35|54.32|44.40|
> |Specialist|49.01|46.99|63.76|57.90|46.34|42.36|66.16|54.78|
>
> We also appreciate that, despite your concerns, you appreciate the soundness of our experimental design and the rigorous ablations which "ensure thorough validation of the effectiveness of same-source ICL specialization". We would also like to address your concerns about our method's novelty and practicality:
> - Novelty: Your review states that "adjusting ICL demonstrations to enhance or analyze the performance of specific tasks has already been extensively studied". This is a broad field of study in its own right, and no supporting citations were provided. The "Essential References Not Discussed" section is also empty. If you believe essential references are missing, we would like to include them, but cannot act on this incomplete comment. Moreover, our contribution is not limited to showing that our proposed method for constructing ICL examples enhances performance of LLM-as-a-Judge metrics. More broadly, our work proposes a novel technique for using human evaluation to improve automatic metrics (rather than using human evaluation as the de facto evaluation method itself) via test set specialization, and includes an extensive study of how rater behavior affects performance of LLM-as-a-Judge metrics prompted with human ratings. Research on how to use human annotations to enhance the quality of automatic metrics is an important research area in its own right. To the best of our knowledge, our Specialist method has never been proposed, and yields enormous gains over the existing SOTA in MT evaluation (54% and 119%, respectively, on WMT'23 and WMT'24 test sets). While we acknowledge the simplicity of the method, its dramatic effectiveness is evident from the results presented in the paper.
> - Generalization to real-world use cases: We would like to address your concern that the WMT benchmark, despite being "standard and widely used", may not "adequately reflect real-world translation scenarios." The submissions to WMT (submitted by researchers from institutions across both academia and industry) include a diverse collection of systems of varying quality, including both LLM-based and non-LLM-based (e.g., small encoder-decoder) systems. As shown in Figure 2 in our paper (and discussed in Section 5.2), the Specialist metric's outperformance is consistent for translation systems across the quality spectrum, and cannot be explained by gains only for a certain quality tier. These results imply that the Specialist metric has *better* generalization to new systems, when removing the constraint that it must also generalize to new test sets.
> - Cost: The review states that "evaluation latency, cost implications, and practicality [...] have not been discussed." Please note that *these considerations are in fact discussed in Section 3 of the paper, in the paragraph entitled "Specialist Method in Practice"*. The review also states that "large-scale deployment of prompted LLM-based metrics can be computationally costly". Once deployed, our proposed Specialist metric does not have any additional computational overhead relative to standard LLM-based metrics. Moreover, large-scale deployment of LLM-based metrics has been widely adopted and, for many real-world applications, is more practical, efficient, and cost-effective than running human evaluations. In fact, "rapid model iteration" is virtually impossible without relying on automatic metrics.
>
> We hope the additions we have made to our paper (especially the new results on the HANNA benchmark for story generation evaluation) allay your concerns about our method's generalizability to other NLG evaluation tasks, and that you will consider increasing our score. Please let us know if you have any further questions or if we can provide any additional clarifications to help finalize your assessment of our paper.

---

### Official Review · Reviewer_tvUC · 2025-03-14

**Overall Recommendation:** 4

**Summary:**

This paper introduces an LLM-based automatic evaluation method, called "Specialist". At a high-level, Specialist closely imitates a human rater's behavior by making ICL examples from the ratings (1) of the same rater (2) on the same example (3) on different model outputs. In other words, the only place where extrapolation from past data points happen is from annotations on past model outputs to new model outputs. Results show that when measured by character-level F1 of the annotated error spans, Specialist outputs the state-of-the-art machine translation metric XCOMEt by 54% and 119% respectively on WMT 23 and WMT 24 test sets.

**Claims And Evidence:**

All claims made in the submission is clearly and convincingly supported by their experimental results.

**Essential References Not Discussed:**

Not that I'm aware of.

**Experimental Designs Or Analyses:**

I carefully checked through all the details in Section 4 and haven't spot any issues.

**Methods And Evaluation Criteria:**

The method makes sense to me, but the authors could make it clearer how this is useful in real-world applications. From my perspective, this is only useful when one has existing human evaluation on a dataset and one is using "Specialist" to obtain accurate, explainable evaluations for future systems developed on this test set. It can't be (1) used on an arbitrary new dataset where no prior human evaluation is available (2) used for alternative QE metric use cases such as data filtering.

The evaluation criteria makes sense, but is constrained to finer granularities such as span-level and segment-level. I have two small doubts, mostly on Section 5.6:

- Why did you use an alternative prompt, instead of directly converting the annotated MQM errors into MQM scores?
- Current evaluation is only constrained to segment-level. I'd like to see how this stacks up to other WMT metrics on system-level on the acc23 dataset. (Specifically, where it stands in Table 1 in the [WMT 24 metrics shared task paper](https://aclanthology.org/2024.wmt-1.2.pdf).)

**Other Comments Or Suggestions:**

- I think this paper shares some resemblance to the spirit of "test-time training" (https://arxiv.org/pdf/1909.13231). The authors may consider drawing an analogy to justify their usage of existing information in the test examples.
- In general, the evaluation is somewhat skewed to error span annotation, while being rather lenient to error classification and severity -- both are actually critical when converting MQM annotations to scores. I wonder why this skew exists. (My comment in "Methods And Evaluation Criteria" is also along that line.)

**Other Strengths And Weaknesses:**

In general, this is a nicely-written paper with good novelty and well applicability to machine translation development, with potentials to extend to evaluations of other generative tasks. The only reservation I have is I'm having trouble understanding Section 5.4.1. The logic in the section is hard to follow, and it looks a little duplicative to 5.4.2 in that both mentioned "if the model is simple copying errors in the ICL examples". Can the authors explain to me what they are trying to achieve in 5.4.1 and what the results mean?

**Questions For Authors:**

The most crucial question is regarding question 5.4.1 (see "Other Strengths And Weaknesses"). Please make sure you address that one.

**Relation To Broader Scientific Literature:**

The proposed Specialist method is a nice addition to the existing LLM-based machine translation evaluation methods. While there are both methods that directly predict scores (GEMBA) as well as ones that predicts MQM annotations (AutoMQM), the contribution of this paper is orthogonal to existing work and concerns about the usage of ICL examples. It can serve as a nice addition to both of these methods in the intended applications.

**Theoretical Claims:**

N/A

---

> ### Author Rebuttal · Authors · 2025-03-31
>
> Dear Reviewer, thank you very much for your positive and thorough review of our paper. We will address your outstanding concerns below, starting with the question which you stated as most important.
> > Can the authors explain to me what they are trying to achieve in 5.4.1 and what the results mean?
>
> Sections 5.4.1 and 5.4.2 both address the question of whether the Specialist model is simply copying errors in the ICL examples, but from two different angles, to provide further evidence for the claim that the success of the Specialist cannot be attributed to naive copying behavior. In Section 5.4.1, we compare copying behavior of prompted Autoraters using Specialist vs non-Specialist (and fewer Specialist) ICL examples, while in Section 5.4.2, we compare the Specialist prompted Autorater with a non-model-based baseline, which makes predictions by directly copying ICL example errors. While the gap in precision between the Specialist Autorater and the Parrot baseline in Section 5.4.2 shows that the Specialist metric is not copying all of the errors that it could from the ICL examples, this alone does not establish that the Specialist ICL examples are actually *teaching the model* which errors to *abstain from predicting*. This latter claim is substantiated in Section 5.4.1, where we contextualize the Specialist's copying behavior with respect to that of other prompted Autoraters, and show that providing *more* Specialist ICL examples results in a *decrease* in copying behavior with respect to these examples. We hope this has clarified the difference between Sections 5.4.1 and 5.4.2, but if anything remains unclear, please let us know.
>
> > Section 5.6: Why did you use an alternative prompt, instead of directly converting the annotated MQM errors into MQM scores?
>
> In Table 5, we are actually directly comparing these two settings. That is, to compute Acc23 for the *AutoMQM Specialist*, we are indeed converting the predicted MQM errors into scores. As shown in the table, this outperforms the *Score Prediction Specialist*, which directly prompts the LLM to generate a float quality score on a scale from 0-100 (aligning with standard direct assessment prompting techniques traditionally used for LLM-as-a-Judge evaluation).
>
> As for the meta-evaluation metrics used, and in particular your concern regarding leniency to error classification and severity, note that the character-level F1 does indeed penalize incorrect error severities. See Section 4.5 of the paper: "Partial credit of 0.5 is given if the predicted rating correctly marks a character as an error but predicts the incorrect severity." To address your concern about leniency towards error classification, we report the "binary error classification F1" meta-evaluation metric below, which computes the F1 of the rating-level binary error vs no-error decision. This complements Table 1 in the paper, and we see that the Specialist metric outperforms all others according to this evaluation too.
> |Binary error class. F1|WMT'23|WMT'24|
> |:----------------------|:-----:|:-----:|
> |XCOMET|83.35|52.36|
> |GEMBA|73.43|-|
> |Shuffled sources|80.25|49.82|
> |Fixed, different source|70.36|49.92|
> |Specialist|89.74|65.23|
>
> > [...] this is only useful when one has existing human evaluation on a dataset and one is using "Specialist" to obtain accurate, explainable evaluations for future systems developed on this test set.
>
> It is true that the Specialist method specializes to a fixed test set by design. By removing the constraint that the Autorater generalize to new test sets, we achieve large gains in the metric's ability to generalize to new systems, and this is often a tradeoff that is worth making. Many benchmarks (e.g., WMT for machine translation, XLSum for Summarization, SQuAD for question answering) are carefully curated and repeatedly used to allow for fair comparison against previous work (including during the model development process). Moreover, the Specialist method provides a paradigm for joint development of *automatic metrics* and *test sets* (which are typically developed independently, but don't need to be, and we get quality gains if they're developed jointly). Thus, when seeking to evaluate on "an arbitrary new dataset where no prior human evaluation is available", the metric development process which accompanies development of this new test set would involve collection of a small set of human annotations of system outputs on this test set.
>
> > [...] resemblance to the spirit of "test-time training" (https://arxiv.org/pdf/1909.13231)
>
> We agree that Test-Time Training has some similarity in spirit with our work, in that we both seek to specialize a model to a particular test set example using data related to that example. While they use related examples from a simpler auxiliary task with automatic labels (image rotation), we use related examples from the same task with ground-truth labels. Additionally, our method does not involve parameter updates.
>
> Thank you!

---

### Official Review · Reviewer_inJi · 2025-03-14

**Overall Recommendation:** 3

**Summary:**

This paper introduces the "Specialist" method, a novel approach to specialize LLM-based Autoraters to specific test sets by leveraging historical ratings through in-context learning (ICL) examples. The method is applied to fine-grained machine translation (MT) evaluation (AutoMQM), achieving significant performance gains—54% and 119% relative F1 improvements on the WMT'23 and WMT'24 test sets respectively, outperforming the state-of-the-art XCOMET metric. The method is robust across different LLM backbones and evaluation tasks, demonstrates sensitivity to rater variability, and suggests potential for broader NLG evaluation applications, though further experimental validation is required.

**Claims And Evidence:**

The claims of superior performance and robustness are convincingly supported by extensive experimentation and thorough comparisons against strong baselines (XCOMET, GEMBA-MQM, MetricX-24). However, the claim regarding the broader applicability to other NLG evaluation tasks lacks empirical validation beyond MT evaluation.

**Essential References Not Discussed:**

The essential reference "BatchEval: Towards Human-like Text Evaluation," which presents a related comparative-based evaluation method, should be discussed to contextualize this paper's contributions further.

**Experimental Designs Or Analyses:**

The experimental design is thorough and sound within the context of MT evaluation. The authors employ strong baselines, rigorous ablations, and analyses to validate the impact of the method, including investigating the effect of ICL example counts, error copying behaviors, robustness across translation systems, and inter-rater variability. Yet, the lack of experimentation on additional NLG tasks limits the overall generalizability claim.

**Methods And Evaluation Criteria:**

The proposed method—constructing ICL examples from historical ratings—is highly suitable for the MT evaluation problem, leveraging existing human annotations effectively. The chosen evaluation criteria (character-level F1 for AutoMQM and Acc23 for scoring tasks) are appropriate and align well with the MT evaluation standards used in the literature. However, additional evaluation across a broader range of NLG tasks (e.g., Topical-Chat, FED, HANNA, QAGS) would strengthen the method's claimed generalization.

**Other Comments Or Suggestions:**

None.

**Other Strengths And Weaknesses:**

Strengths:

1. The paper is well written.

2. Innovative and simple approach with practical significance.

3. Extensive empirical validation and robustness analyses.

4. Clearly articulated results and experimental methodology.

Weaknesses:

1. The method's reliance on historical annotations might limit applicability in settings without existing annotation resources.

2. Generalization to broader NLG evaluation tasks remains experimentally unverified.

3. Absence of a detailed comparative discussion with the BatchEval method, which uses similar comparative-based ideas.

**Questions For Authors:**

Questions For Authors*

Q1: Could you clarify how their method relates to, differs from, or improves upon BatchEval: Towards Human-like Text Evaluation?

Q2: Could you validate the method across other widely-used NLG evaluation tasks such as Topical-Chat, FED, HANNA, or QAGS?

**Relation To Broader Scientific Literature:**

The paper builds effectively on the existing literature around automatic MT evaluation, specifically extending the LLM-as-a-Judge paradigm. It directly relates to and advances beyond GEMBA-MQM and XCOMET methods by introducing specialization via ICL examples from historical evaluations. However, it lacks discussion of the relation, similarities, and differences to the closely related BatchEval method (BatchEval: Towards Human-like Text Evaluation), which also employs a comparative evaluation framework.

**Theoretical Claims:**

No explicit theoretical proofs or claims are provided or necessary for this empirical study. The focus is empirical validation rather than theoretical contributions.

---

> ### Author Rebuttal · Authors · 2025-03-31
>
> Dear Reviewer, thank you very much for your thoughtful and detailed review of our paper. We will address your concerns below.
> >*Q1*: Could you clarify how their method relates to, differs from, or improves upon BatchEval: Towards Human-like Text Evaluation? (same as *Weakness 3*)
>
> BatchEval is only related to our work through our use of in-context learning (ICL), as both approaches involve exposing the model to predictions from either itself (BatchEval) or the ground truth (Specialist) on different examples. BatchEval exposes the model to its own predictions on a batch of unrelated other samples. In this work, we expose the model to ground-truth predictions on a batch of related (same-source) samples. The approaches are not in competition with each other: one could provide multiple same-source historical annotations as ICL examples (our work) while asking the evaluator model to evaluate a batch of different outputs from the same source in one or more stages (BatchEval). We will add this BatchEval reference to the Related Work section in the final version of our paper.
> > *Weakness 1*: The method's reliance on historical annotations might limit applicability in settings without existing annotation resources.
>
> It is true that access to (an offline collection of) human annotations is inherent to our work, which proposes a novel technique for using human evaluation to improve automatic metrics (rather than using human evaluation as the de facto evaluation method itself) via test set specialization. This work contributes to research on how to use human annotations to enhance the quality of automatic metrics, which we believe is an important and under-explored field, and our hope is that the results in our paper showing the effectiveness of the Specialist method will provide motivation to the broader NLG community to acquire human annotations and build Specialist metrics for other tasks and datasets.
>
> > *Q2*: Could you validate the method across other widely-used NLG evaluation tasks such as Topical-Chat, FED, HANNA, or QAGS?
> (same as *Weakness 1*)
>
> Thank you for pointing us to a suitable non-MT NLG evaluation dataset, which has enabled us to improve our paper by validating the generalization of our method. We have added a new experiment showing the effectiveness of the Specialist method on the HANNA benchmark (https://arxiv.org/pdf/2208.11646; one of your recommendations), which evaluates story generation. Some additional details of our experimental setup are below:
> - We evaluated both with and without human stories included. The HANNA paper excluded human stories, as these were considered outliers, while some follow-up papers included them.
> - The HANNA dataset includes 6 criteria: relevance, coherence, empathy, surprise, engagement, and complexity. Due to time constraints, we report results on the first criterion (Relevance) and on the average over all criteria (where the averaged score represents an overall indication of the story's quality).
> - We follow the same experimental methodology as described in Section 4.4 of our paper, and construct Specialist ICL examples per each test example via hold-one-system-out prompting. This simulates the real-world use case of evaluating a new system.
>
> As shown in the table below (which reports story-level Pearson and Spearman correlations), our Specialist method demonstrated substantial gains on HANNA, outperforming the shuffled baseline across both tasks and all meta-evaluation metrics. On the Relevance task (REL), the Specialist achieved a Pearson of 49.0 (excl. human stories), compared to 43.7 for the shuffled baseline, which already outperformed the winning metric (BARTScore; Pearson=42.6) reported in the [HANNA paper](https://arxiv.org/pdf/2208.11646). The gap in performance between the Specialist and shuffled baseline is even wider when evaluating on the average score, with Specialist Pearson of 46.3 (66.2 incl. human stories), compared to 36.7 (54.3 incl. human stories) for the shuffled baseline.
> |Metric|REL (excl. human)||REL (incl. human)| |AVG (excl. human)| |AVG (incl. human)| |
> |:-----------|:-------:|:-------:|:-------:|:-------:|:-------:|:-------:|:-------:|:-------:|
> ||r_p|r_s|r_p|r_s|r_p|r_s|r_p|r_s|
> |Shuffled baseline|43.71|41.51|59.58|52.88|36.74|32.35|54.32|44.40|
> |Specialist|49.01|46.99|63.76|57.90|46.34|42.36|66.16|54.78|
>
> We believe this new experiment has strengthened our paper substantially by showing that the Specialist method generalizes to a long-form text generation evaluation task (with very different characteristics than machine translation evaluation).
>
> We hope the additions we have made to our paper (especially the results on the HANNA benchmark and the comparison with the BatchEval method) allay your concerns and will be taken into account when assigning your final score. Please let us know if you have any further questions or if we can provide any additional clarifications to help finalize your assessment of our paper.

---

### Decision · Program_Chairs · 2025-05-01

**Decision:**

Accept (poster)

**Comment:**

This is an empirical paper with a clear contribution to the growing literature on LLM-based evaluation metrics. While the technique is relatively simple, its demonstrated effectiveness on key benchmarks  justifies its contribution.

There are some concerns raised around generalizability and practicality of the method.
The majority of reviewers, however, found the work valuable and sound, and recommend acceptance.